

# Terrestrial exospheric dayside H-density profile at 3-15 $R_{\mathrm{e}}$ from UVIS/HDAC and TWINS Lyman-$\alpha$ data combined

Jochen H. Zoennchen[1], Hyunju K. Connor[2], Jaewoong Jung[2], Uwe Nass[1], and Hans J. Fahr[1]

[1]Argelander Institut für Astronomie, Astrophysics Department, University of Bonn, Auf dem Huegel 71,
53121 Bonn, Germany

[2]Geophysical Institute, University of Alaska Fairbanks, Alaska, USA

*Correspondence to:* J. H. Zoennchen (zoenn@astro.uni-bonn.de)

**Abstract.**
Terrestrial ecliptic dayside observations of the exospheric Lyman-$\alpha$ column intensity between 3-
15 Earth radii ($R_{\mathrm{e}}$) by UVIS/HDAC at CASSINI have been analysed to derive the neutral exospheric
H-density profile at the Earth's ecliptic dayside in this radial range. The data were measured during
CASSINIS's swing by manoeuvre at the Earth on 18 August 1999 and are published by (Werner
et al., 2004). In this study the dayside HDAC Lyman-$\alpha$ observations published by (Werner et al.,
2004) are compared to calculated Lyman-$\alpha$ intensities based on the 3D H-density model derived
from TWINS Lyman-$\alpha$ observations between 2008-2010 (Zoennchen et al., 2015). It was found, that
both Lyman-$\alpha$ profiles show a very similar radial dependence in particular between 3-8 $R_{\mathrm{e}}$. Between
3.0-5.5 $R_{\mathrm{e}}$ impact distance Lyman-$\alpha$ observations of both TWINS and UVIS/HDAC are existing at
the ecliptic dayside. In this overlapping region the cross-calibration of the HDAC profile against
the calculated TWINS profile was done, assuming, that the exosphere there was similar for both
due to comparable space weather conditions. As result of the cross-calibration the conversion factor
between counts/s and Rayleigh $f_c$=3.285 [counts/s/R] is determined for these HDAC observations.
Using this factor the radial H-density profile for the Earths ecliptic dayside was derived from the
UVIS/HDAC observations, which constrained the neutral H-density there at 10 $R_{\mathrm{e}}$ to a value of 35
$cm^{-3}$. Furthermore, a faster radial H-density decrease was found at distances above 8 $R_{\mathrm{e}}$ ($\approx r^{-3}$)
compared to the lower distances 3-7 $R_{\mathrm{e}}$ ($\approx r^{-2.37}$). This increased loss of neutral H above 8 $R_{\mathrm{e}}$
might indicate a higher rate of H ionization in the vicinity of the magnetopause at 9-11 $R_{\mathrm{e}}$ (near sub
solar point) and beyond, because of increasing charge exchange interactions of exospheric H atoms
with solar wind ions outside the magnetosphere.



**Keywords.** Atmospheric composition and structure (airglow and aurora; pressure, density, and tem-
perature) – meteorology and atmospheric dynamics (thermospheric dynamics)

## 1 Introduction

The Earth's exosphere is the outermost layer of our atmosphere that ranges from ≈500km altitude
to beyond the Moons orbit (Baliukin et al. 2019). Atomic hydrogen atom (H) becomes a dominant
species above an altitude of ≈1500km. The exosphere gains and loses hydrogen atoms as a result of
the Sun - solar wind - magnetosphere - upper atmosphere interaction. Study of the exospheric density
distribution and its response to dynamic space environments is key to understand the past, present,
and future of the Earths atmosphere and to infer the evolution of other planetary atmospheres.
The typical geocorona emission, i.e., solar Lyman-$\alpha$ photons resonantly scattered by hydrogen
atoms, has been a widely used dataset to derive a terrestrial exospheric neutral H-density. Sev-
eral spacecraft missions like Thermosphere - Ionosphere - Mesosphere Energetics and Dynamics
(TIMED; Kusnierkiewicz, 1997), Two Wide-Angle Imaging Neutral- Atom Spectrometer (TWINS;
Goldstein & McComas, 2018), and Solar and Heliospheric Observatory (SOHO; Domingo et al.,
1995) have observed the geocorona from various vantage points, covering an optically thick, near-
Earth exosphere below ≈3 $R_e$ geocentric distance (e.g., Qin & Waldrop, 2016; Qin et al., 2017; Wal-
drop et al., 2013) to an optically thin, far distant exosphere on top (e.g., Bailey & Gruntman, 2011;
Cucho-Padin & Waldrop, 2019; Zoennchen et al., 2011, 2013). The exospheric density changes over
various time scales such as solar cycle (Waldrop & Paxton, 2013; Zoennchen et al., 2015; Baliukin
et al., 2019), solar rotation (Zoennchen et al., 2015), and geomagnetic storms (Bailey & Gruntmann,
2013; Cucho-Padin & Waldrop, 2019; Qin et al., 2017; Zoennchen et al., 2017). This implies active
response of our exosphere to a dynamic space environment through physical processes like thermal
expansion, photoionization, and neutral charge exchanges as suggested in the previous theoretical
studies (Chamberlain, 1963; Bishop, 1985; Hodges, 1994; and references therein). Also the possible
contribution of non-thermal hydrogen to the exosphere is discussed (e.g., Qin & Waldrop, 2016;
Fahr et al., 2018).
Recently, exospheric neutral H-density at 10 $R_e$ subsolar location becomes a particular interest
due to two upcoming missions, the NASA Lunar Environment heliospheric X-ray Imager (LEXI;
http://sites.bu.edu/lexi) and the joint ESA-China mission, Solar wind - Magnetosphere - Ionosphere
Link Explorer (SMILE; Branduardi-Raymont et al., 2018) with expected launches in 2023 and 2024,
respectively. Soft X-ray imagers on these spacecrafts will observe motion of the Earths magne-
tosheath and cusps in soft X-ray with a primary goal of understanding the magnetopause reconnec-
tion modes under various solar wind conditions. Soft X-ray is emitted due to interaction between the
exospheric neutrals and the highly charged solar wind ions like $O^{7+}$ and $O^{8+}$ (Sibeck et al., 2018;
Connor et al., 2021). Neutral density is a key parameter that controls the strength of soft X-ray sig-



nals. Denser hydrogen increase their interaction probability with solar wind ions and thus enhance
soft X-ray signals, which is preferable for the LEXI and SMILE missions.
The dayside geocoronal observations above 8 $R_e$ radial distance are very rare. For estimating an
exospheric density at 10 $R_e$ subsolar location, Connor & Carter (2019) and Fuselier et al. (2010;
2020) used alternative datasets: the soft X-ray observations from the X-ray Multi-Mirror Mission-
Newton astrophysics mission (XMM; Jansen et al., 2001) and the Energetic Neutral Atom (ENA)
observations from the Interstellar Boundary Explorer (IBEX; McComas et al., 2009), respectively.
Their density estimates at 10 $R_e$ show a large discrepancy, ranging from 4 $cm^{-3}$ to 59 $cm^{-3}$ with a
lower limit from the IBEX observations and an upper limit from the XMM observations. However,
these studies analyzed only a handful of events. Additionally, inherent difference of the soft X-ray
and ENA datasets leads to different density extraction techniques, possibly contributing to the neu-
tral density discrepancy. To understand a true nature of this outer dayside exosphere, more statistical
and cumulative approaches with various datasets are needed.
We estimate a dayside exospheric density in a radial distance of 3-15 $R_e$ using rare dayside geo-
corona observations obtained from the CASSINI UVIS/HDAC Lyman-$\alpha$ instrument on 18 August
1999. This paper is structured as follows. Section 2 introduces the CASSINI Lyman-$\alpha$ observations
on 18 August 1999. Section 3 discusses the solar condition and interplanetary Lyman-$\alpha$ background
during the observation period. Section 4 explains our density extraction approach. Section 5 esti-
mates the conversion factor of the CASSINI UVIS/HDAC geocorona count rates to Rayleigh, and
Section 6 derives the dayside exospheric density profiles from the converted geocoronal emission in
Rayleigh. Finally, Section 7 discusses and concludes our results.

## 79  2   The UVIS/HDAC Lyman-$\alpha$ observations during CASSINIs swing by at the Earth

On its way to Saturn the CASSINI spacecraft performed a swing by manoeuvre at the Earth on 18
August 1999. The UVIS/HDAC Lyman-$\alpha$ instrument (FOV $\approx 3°$) was switched on before and mea-
sured then continuosly Lyman-$\alpha$ intensities during the manoeuvre. When approaching the Earth the
measured Lyman-$\alpha$ intensities were increasingly dominated by scattered Lyman-$\alpha$ emission from
neutral H-atoms of the terrestrial exosphere. The intensity profile in [counts/s] (averaged over a
1 min interval) from UVIS/HDAC is a rare observation of the exospheric dayside Lyman-$\alpha$ emis-
sion near the Earth-Sun line up to 15 $R_e$ geocentric distance. It is a nearly perfect scan within the
ecliptic plane during $\approx 1.5$ hours and therefore nearly free from latitudinal and temporal variations.
The profile was published by (Werner et al., 2004) and is shown in Figure 2 of their paper. From
each measurement they had subtracted 4500 [counts/s] as correction for their estimate of the inter-
planetary background intensity. For the geocentric distances 3-15 $R_e$ this corrected profile can be





numerically approximated by the following fit function:
$$I_{corr}(r) = 282920.2 * (r + 2.0)^{-2.2} \ [counts/s] \tag{1}$$
with the geocentric distance r in $R_e$. In Figure 1 is shown, that the fitted radial intensity function
from Equation (1) (red line) approximates the profile from (Werner et al., 2004) (black line) very
well. Values from Equation (1) need to be re-added with 4500 [counts/s] in order to retrieve the
uncorrected intensities originally measured by UVIS/HDAC:
$$I(r) = I_{corr}(r) + 4500 \ [counts/s] \tag{2}$$
The observational geometry (spacecraft position and viewing direction of UVIS/HDAC) during the
swing by was also adopted from (Werner et al., 2004): On the Earth dayside CASSINI moved within
the ecliptic plane towards Earth. CASSINI's dayside trajectory as shown in (Werner et al., 2004 -
see Figure 1 there) is nearly linear within 3-15 $R_e$. It can be numerically approximated as radial
function of the GSE longitude:
$$\phi_{GSE}(r) = 6.7 + 80.14/r \ [°] \tag{3}$$
with the geocentric distance r in $R_e$. Following (Werner et al., 2004) in this trajectory segment the
line of sight (LOS) of UVIS/HDAC pointed towards the positive GSE Y-axis away from Earth.
**3 Solar conditions and the interplanetary Lyman-$\alpha$ background**
On the swing by date 18 August 1999 the value of the total solar Lyman-$\alpha$ flux was $4.52 \cdot 10^{11}$
[photons/cm$^2$/s]. It has been mesured by TIMED SEE and SORCE SOLSTICE calibrated to UARS
SOLSTICE level [Woods et al., 2000] (provided by LASP, Laboratory For Atmospheric And Space
Physics, University of Boulder, Colorado). With the function given by (Emerich et. al., 2005), the
line-center solar Lyman-$\alpha$ flux was calculated from this total solar Lyman-$\alpha$ flux for the derivation
of the g-factor as used in Equation (4).
The solar activity level as indicated by the solar $F_{10,7cm}$-radio flux starts to increase in summer 1999
from the low values of the solar minimum until 1998.
When flying at the Earth dayside between 3-15 $R_e$ the UVIS/HDAC LOS pointed to a region with in-
terplanetary Lyman-$\alpha$ background of about 1400 $R$. This value was taken from the SOHO-SWAN all
sky map of the Lyman-$\alpha$ background of 17 August 1999 (SOHO-SWAN images provided via Web by
LATMOS-IPSL, Universit Versailles St-Quentin, CNRS, France: http://swan.projet.latmos.ipsl.fr/images/).
**4 Approach**
During the swing by at the Earth the UVIS/HDAC instrument measured Lyman-$\alpha$ radiation reso-
nantly backscattered from neutral hydrogen of the terrestrial exosphere and also from the interplan-
etary medium. Due to their low velocities the contributing H-atoms can be considered as "cold".





Therefore, this backscattered radiation contains wavelengths with a relatively narrow bandwidth
around the Lyman-$\alpha$ line center. The sole contribution of the interplanetary hydrogen was quantified
by the value taken from SOHO-SWAN as described in the previous section.
Within the exosphere the optical depth turns to be lower than 1 at geocentric distances $> 3\ R_e$,
which allows for the assumption of single scattering. Under this assumption for a particular solar
Lyman-$\alpha$ radiation (manifested in the g-factor) the exospheric H-density $N(S)$ along a line of sight
$S$ produces a Lyman-$\alpha$ scatter intensity $I$ in [R]:
$$I = \frac{g}{10^6} \int_0^{S_{max}} n(S)\epsilon(S)I_p(\alpha(S))dS \tag{4}$$

with $n(S)$ is the local H-density, $\epsilon(S)$ the local correction term for geocoronal selfabsorption/re-
emission and $I_p(\alpha(S))$ the local intensity correction for the angular dependence of the scattering.
Additionally to the solar radiation the dayside Lyman-$\alpha$ observations above $3R_\mathrm{e}$ analysed in this
study are illuminated by a secondary Lyman-$\alpha$ radiation from lower atmospheric shells of the Earth:
At the dayside lower, optically thick exospheric shells are face-on illuminated by the Sun. The re-
emission created there acts as a secondary source of Lyman-$\alpha$ besides the Sun. The relative effect
increases with decreasing geocentric distance. With the $\epsilon(S)$-term in Equation (4) the Lyman-$\alpha$ in-
tensity profile can be corrected from re-emission of solar Lyman-$\alpha$ from lower atmospheric shells of
the Earth. The applied method in this study, all considered correction terms and the used $\epsilon(r, \theta, \phi)$
map (shown in Figure 2) are in detail described in (Zoennchen et al., 2015).
With usage of a given H-density distribution the Lyman-$\alpha$ column brightness can be calculated for
any LOS and observing position within the optically thin regime based on the integral in Equation
(4). The calculated values ([R]) can be converted into their observable intensities ([counts/s]) using
a single instrumental factor ([counts/s/R]) - further refered as conversion factor $f_c$.
In this study two H-density models are used for comparison with UVIS/HDAC: the exospheric
$H(r, \theta, \phi)$-density model derived from TWINS Lyman-$\alpha$ observations from 2008 and 2010 (Zoen-
nchen et al., 2015) and a radial symmetric model as introduced by (Chamberlain, 1963) and fre-
quently used for example by (Rairden et al., 1986), (Fuselier et al., 2010, 2020) or (Connor &
Carter, 2019):
$$n_H(r) = n_0 \cdot \left( \frac{10\ R_e}{r} \right)^3 \tag{5}$$

with the geocentric distance r in $R_\mathrm{e}$. The H-density at 10 $R_\mathrm{e}$ subsolar point ($n_0$) is set at 40 $cm^{-3}$,
which is within the reported range of Connor & Carter (2019) that derived $n_0$ from the XMM soft
X-ray emission.
The comparison of the calculated profiles with the UVIS/HDAC profile was made for two reasons:
First, to compare their radial dependency and second, to derive the conversion factor $f_c$ of UVIS/HDAC
by cross-calibrating it against the calculated profile from the TWINS H-density model in the radial
range 3.0-5.5 $R_\mathrm{e}$ (overlapping range). Dayside Lyman-$\alpha$ observations with impact distances inside


this overlapping range are available by both - UVIS/HDAC and TWINS. This method for evalua-
tion of $f_c$ assumes, that the TWINS H-density model from 2008, 2010 also matches the exospheric
H-density distribution on 18 August 1999 due to comparable space weather conditions. Both, the
used TWINS and UVIS/HDAC observations were measured during quiet geomagnetic conditions
(minimum Dst index $\approx$ -30 nT; provided by the website of the WDC for Geomagnetism, Kyoto) and
low solar activity (Solar 10,7 cm $\leq$ 130).
Nevertheless, it is known from other studies, that the terrestrial exosphere show H-density variations
of about 10-20% caused by geomagnetic storms (i.e. Bailey & Gruntman, 2013; Zoennchen et al.,
2017; Cucho-Padin & Waldrop, 2018). Therefore we expect an error of the conversion factor by this
variations up to 20%.

## 5   Comparison of the observed UVIS/HDAC profile with calculated profiles

The observed dayside Lyman-$\alpha$ profile (column intensity) by UVIS/HDAC (approximated in Equa-
tion (2)) was compared to the calculated Lyman-$\alpha$ profiles (column brightness) from two exospheric
H-density models described in the previous section. CASSINI's trajectory at the dayside between
3-15 $R_{\mathrm{e}}$, the LOS of HDAC, the interplanetary background and the solar conditions of the swing by
day 18 August 1999 were considered by the calculation.
Figure (3A) shows the uncorrected observed Lyman-$\alpha$ profile by UVIS/HDAC from Equation (2)
in [counts/s] (black line) together with the calculated column brightness profiles in [R] based on the
TWINS 3D H-density model (red line) and the $1/R^3$ model (blue line) - all including interplanetary
Lyman-$\alpha$ background. It is obvious from that figure, that between 3-8 $R_{\mathrm{e}}$ the radial dependence of
the calculated profile using the TWINS 3D H-density model corresponds well to the UVIS/HDAC
observed profile. The radial dependency of the $1/R^3$-profile (blue line) deviates from the HDAC
profile in this particular range.
Figure (3B) shows the ratios of the observed and the calculated profiles: In the overlapping range
(3.0-5.5 $R_{\mathrm{e}}$) the averaged ratio between the UVIS/HDAC observations and the TWINS 3D H-density
model (red line) is nearly constant with only slight variations between -2.1% and +1.2%. It is equiv-
alent to the averaged conversion factor and was found to be $f_c$=3.285 [counts/s/R].
For the $1/R^3$ model (blue line) the ratio shows significant deviations from a constant value for lower
radial distances $<8R_{\mathrm{e}}$. But for distances above 9 $R_{\mathrm{e}}$ the profile of this model turned also into a
nearly constant ratio to the UVIS/HDAC data (average = 3.145 [counts/s/R]).

Besides the cross-calibration method there is another independent way to approximate $f_c$: (Werner
et al, 2004) estimated the interplanetary Lyman-$\alpha$ background in the UVIS/HDAC observations with
4500 [counts/s]. To be not contaminated with exospheric emission, this value had to be measured
far enough outside the exosphere. The interplanetary Lyman-$\alpha$ radiation is also created by resonant





backscattering and is therefore comparable in its physical properties to exospheric emission. Using
the Lyman-$\alpha$ background emission value from SOHO-SWAN in [R] for the UVIS/HDAC LOS, the
conversion factor $f_c$ can be approximated on this separate way to:
$$f_c = \frac{4500 \ counts/s}{1400 \ R} = 3.215 \ [counts/s/R] \tag{6}$$
The two results for $f_c$ with $f_c$=3.285 from the profile comparison using the TWINS H-density model
and $f_c$=3.215 from the background estimation by (Werner et al., 2004) are relatively close together.

## 6   H-density profile derived from the UVIS/HDAC observations

We applied the determined conversion factor $f_c$=3.285 [counts/s/R] to convert the observed dayside
Lyman-$\alpha$ profile by UVIS/HDAC from intensities [counts/s] into column brightness [R] between
3-15 $R_\mathrm{e}$. Inverse usage of Equation (4) with known column brightnesses I(S) allows to fit the H-
density profile. The H-density profile inverted from the UVIS/HDAC observations was fitted into
the radial symmetric function:
$$n_H(r) = 370520 * (r + 2.47)^{-3.67} \ [cm^{-3}] \tag{7}$$
with geocentric distance r in $R_\mathrm{e}$. Figure (4) shows the fitted H-density profile (black line). From the
$n_H(r)$-profile the UVIS/HDAC observations can be calculated very precisely over the entire radial
range 3-15 $R_\mathrm{e}$ within $\pm$ 2% error.
Obvious in Figure (4) is a change in the radial dependency of the profile in the radial region above
8 $R_\mathrm{e}$. At distances lower 8 $R_\mathrm{e}$ the H-density profile seems to fall with distance with a power law $\approx$
$r^{-2.37}$ (red line in Figure (4)). It was fitted in the distance range 3-7 $R_\mathrm{e}$ to:
$$n_H(r) = 10198 * r^{-2.375} \ [cm^{-3}] \tag{8}$$
where the geocentric distance r is in $R_\mathrm{e}$. The black and red lines are in very good agreement at
3-7 $R_\mathrm{e}$. Above >8 $R_\mathrm{e}$ the situation has changed and the H-density falls with about $\approx r^{-3}$, what is
indicated by the very good agreement of the cyan with the black line there. The fit of the H-density
profile between 9-15 $R_\mathrm{e}$ delivers a $r^{-3}$ fall:
$$n_H(r) = 35.17 * \left( \frac{10 \ R_e}{r} \right)^{3.02} \ [cm^{-3}] \tag{9}$$
From theory an enhanced loss of neutral H atoms near the magnetopause and outside the magne-
tosphere can be expected due to sharply increased interactions with solar wind ions in this region
that produces soft X-ray photons and ENAs. The faster decrease with $r^{-3}$ in the H-density profile
above 8 $R_\mathrm{e}$ might indicate the higher ionization of cold exospheric neutrals near the magnetopause
(located at 9-11 $R_\mathrm{e}$ in the vicinity of the sub solar point) and beyond.
From the fitted H-density profile of Equation (7) the exospheric H-density at 10 $R_\mathrm{e}$ was found to be
35 $cm^{-3}$ at the ecliptic dayside. From known variations of the neutral exosphere due to geomagnetic





storms up to 20 % and with the summarized error from other contributions (i.e. from background,
solar Lyman-$\alpha$ flux and so on) there is a total error in the H-density of about 25 % expectable. Never-
theless, from several facts we assume, that the found value of 35 $cm^{-3}$ at 10 $R_\mathrm{e}$ is more likely to be
a lower limit: First, between 3-10 $R_\mathrm{e}$ the neutral exospheric response to geomagnetic storms is so far
known as an increase and not as a decrase of neutral density (Bailey & Gruntman 2013, Zoennchen
et al., 2017, Cucho-Padin & Waldrop 2018). Second, there are results from other studies, that an
increasing solar activity also corresponds to an increase of neutral density in this radial range, either
weak (Fuselier et al., 2020) or somewhat stronger (Zoennchen et al., 2015). The H-density model
from TWINS used here based on observations in 2008 and 2010 near solar minimum during quiet
days without storms. Therefore it represents likely an exosphere with neutral densities close to their
lowest values.

## 7 Discussion

Ecliptic dayside Lyman-$\alpha$ observations of the terrestrial H-exosphere between 3-15 $R_\mathrm{e}$ by UVIS/HDAC
onboard CASSINI were compared to calculated Lyman-$\alpha$ brightnesses using two different H-density
models: First, the H-density model based on TWINS Lyman-$\alpha$ observations from 2008, 2010 and
second, the $1/R^3$-model introduced by (Chamberlain et al., 1963). The calculations considered the
HDAC Lyman-$\alpha$ observations, CASSINIs trajectory and the HDAC LOS published by (Werner et
al., 2004).
As first result it was found, that the radial dependence of the HDAC observations and the calculated
profile from the TWINS model are very similar, in particular in the radial range 3-8 $R_\mathrm{e}$. The Cham-
berlain model shows significant deviations from the observed profile in this lower range.
To be able to convert the HDAC observations from [counts/s] into physical units [R] the averaged
conversion factor $f_c$=3.285 [counts/s/R] was derived in the radial range 3.0-5.5 $R_\mathrm{e}$ (overlapping re-
gion) from the ratio between the HDAC observations and the calculated Lyman-$\alpha$ brightnesses from
the TWINS model. Dayside LOSs with impact distances in the overlapping region are available
from both instruments - HDAC and TWINS LAD. Additionally a second independent way was used
to quantifiy the conversion factor $f_c$=3.215 [counts/s/R] by calculating the ratio between the esti-
mated background value given by (Werner et al., 2004) and the corresponding value taken from the
SOHO/SWAN map. Both values found for $f_c$ are very close together.
With usage of $f_c$=3.285 the HDAC observations are inverted into a radial symmetric H-density pro-
file of the ecliptic dayside between 3-15 $R_\mathrm{e}$. The derived density profile determined a H-density
value of 35 $cm^{-3}$ at 10 $R_\mathrm{e}$ in the vicinity of the sub-solar point. The error is expected with 25 %.
Nevertheless, from different mentioned reasons it is more likely, that this value is closer to the lower
limit.
Also found was a faster decrease of the H-density for distances above 8 $R_\mathrm{e}$ ($r^{-3}$) compared to the

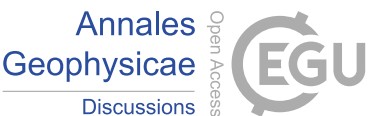

lower region 3-7 $R_e$ ($r^{-2.37}$). This is consistent with an enhanced depletion of neutral H in the far-
upsun direction beyond 8 $R_e$ reported by (Carruthers et al, 1976) based on Lyman-$\alpha$ images from
the Moon by Apollo 16 and also with observations of Mariner 5 (Wallace et al., 1970).
The faster H-density decrease above 8 $R_e$ in the up-sun direction as quantified in this study may
indicate an enhanced ionization rate near the magnetopause and beyond, respectively, due to sharply
increased interactions there of neutral H atoms with solar wind ions.
*Acknowledgements.* The authors gratefully thank the TWINS team (PI Dave McComas) for making this work
possible. Hyunju K. Connor gratefully acknowledges support from the NSF grants, AGS-1928883 and OIA-
1920965, and the NASA grants, 80NSSC18K1042, 80NSSC18K1043, 80NSSC19K0844, 80NSSC20K1670,
and 80MSFC20C0019. We acknowledge the support from the International Space Science Institute on the ISSI
team 492, titled "The Earth's Exosphere and its Response to Space Weather".



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





**Figure Captions**

**Fig. 1.** Black: UVIS/HDAC Lyman-$\alpha$ intensity profile [counts/s] (black line) from (Werner et al., 2004); the origin of the two peaks were identified by (Werner et al., 2004) as (A) the Earthmoon and (B) distortion by the radiation belt; Red: numerical approximation of the intensity profile from Equation (1).

**Fig. 2.** Local ratio $\epsilon(r, \theta, \phi)$ of the local Lyman-$\alpha$ illumination (influenced by multiple scattering effects) and the original solar illumination within the ecliptic plane calculated with a multiple scattering Monte Carlo model (Zoennchen et al., 2015).

**Fig. 3.** (A) observed, uncorrected Lyman-$\alpha$ profile by UVIS/HDAC in [counts/s] from Equation (2) (black line) and the calculated column brightness profiles based on the TWINS 3D H-density model (red line) and the $1/R^3$ model (blue line), both including background and in [R]
(B) ratios between the UVIS/HDAC observed and the calculated profiles: with the TWINS H-density model (red line) and with the $1/R^3$ model (blue line).

**Fig. 4.** (Black line): Radial symmetric H-density profile (Equation (7)) fitted from UVIS/HDAC observations; (Red line): Powerlaw fit of the H-density profile in the lower radial range 3-7 $R_{\mathrm{e}}$; (Cyan line): Powerlaw fit of the H-density profile in the upper radial range 9-15 $R_{\mathrm{e}}$; The deviation of the red and the cyan lines from the black line indicate, that the H-density profiles falls faster at larger distances $>8$ $R_{\mathrm{e}}$ than at lower distances $<8$ $R_{\mathrm{e}}$.





**Figures:**

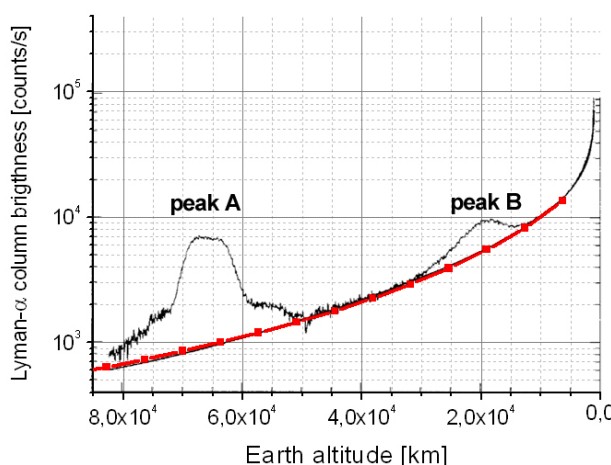

**Fig. 1.** Black: UVIS/HDAC Lyman-α intensity profile [counts/s] (black line) from (Werner et al., 2004); the origin of the two peaks were identified by (Werner et al., 2004) as (A) the Earthmoon and (B) distortion by the radiation belt; Red: numerical approximation of the intensity profile from Equation (1)

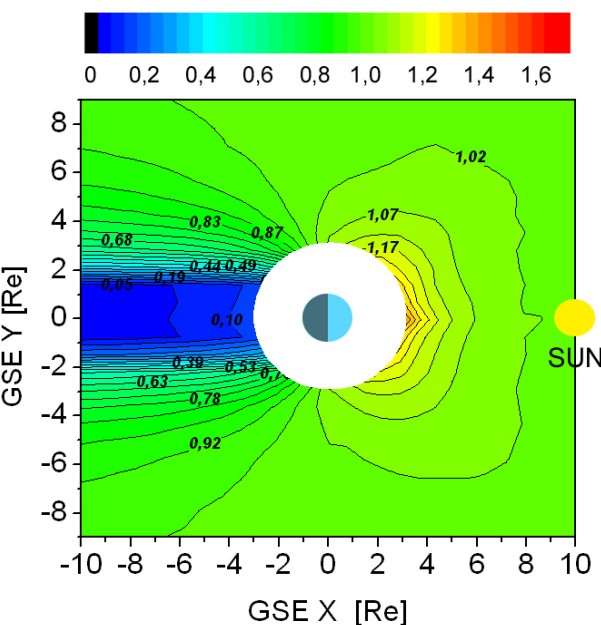

**Fig. 2.** Local ratio ε(r,θ,φ) of the local Lyman-α illumination (influenced by multiple scattering effects) and the original solar illumination within the ecliptic plane calculated with a multiple scattering Monte Carlo model (Zoennchen et al., 2015)



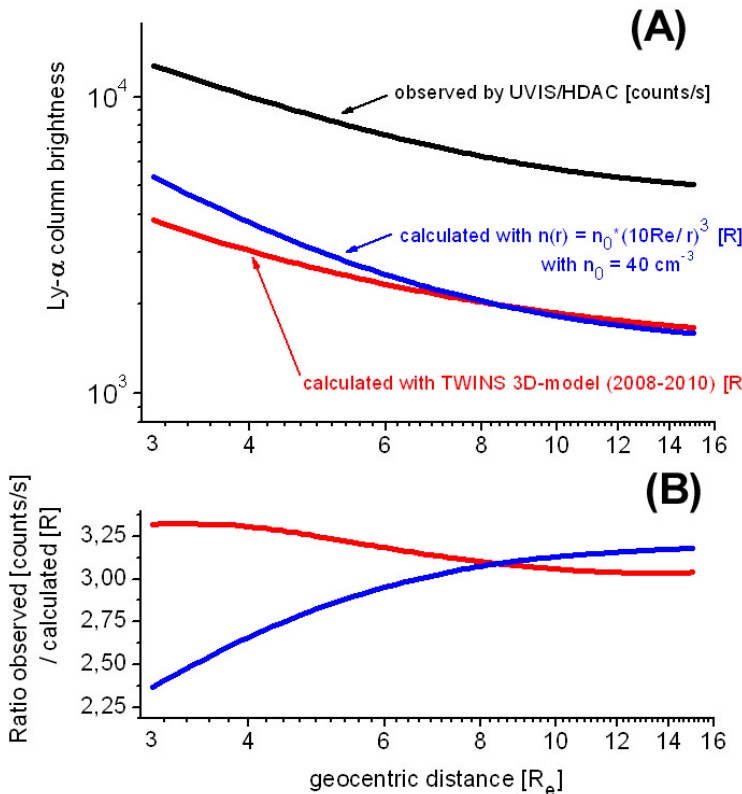

**Fig. 3.** (A) observed, uncorrected Lyman-α profile by UVIS/HDAC in [counts/s] from
Equation (2) (black line) and the calculated column brightness profiles based on the TWINS
3D H-density model (red line) and the $1/R^3$ model (blue line), both including background and
in [R]
(B) ratios between the UVIS/HDAC observed and the calculated profiles: with the TWINS H-
density model (red line) and with the $1/R^3$ model (blue line).



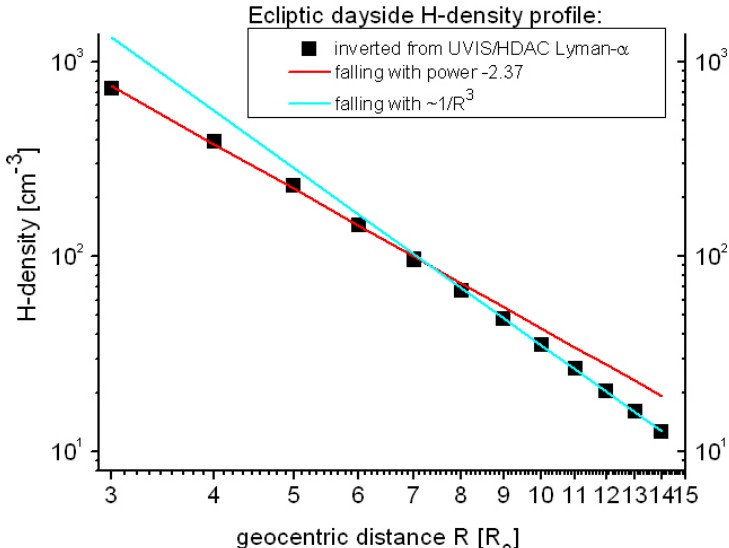

**Fig. 4.** (Black line): Radial symmetric H-density profile (Equation (7)) fitted from UVIS/HDAC observations; (Red line): Powerlaw fit of the H-density profile in the lower radial range 3-7 $R_e$; (Cyan line): Powerlaw fit of the H-density profile in the upper radial range 9-15 $R_e$; The deviation of the red and the cyan lines from the black line indicate, that the H-density profiles falls faster at larger distances >8 $R_e$ than at lower distances <8 $R_e$.