# Peer review of "Terrestrial exospheric dayside H-density profile at 3-15 $R_e$ from UVIS/HDAC and TWINS Lyman- $\alpha$ data combined"

_Annales Geophysicae, 2021_

## Referee Comment (RC1)

Review: *Terrestrial exospheric dayside H-density profile at 3-15 Re from UVIS/HDAC and TWINS Lyman-α data combined*
Jochen H. Zoennchen1, Hyunju K. Connor2, Jaewoong Jung2, Uwe Nass1, and
Hans J. Fahr1

Referee: J. D. Perez

General Comments: This paper describes original analysis of interesting data and should be published.  The challenge of obtaining quantitative information regarding the exospheric dayside H-density profile is illustrated.  Nevertheless as noted in the next section of this Review, a more complete description of the models and the theoretical explanations of the differences would more effectively communicate the significance of the results..

Scientific Questions/Issues:
Line 18-21 and Line 263-265: Authors suggest that the faster radial H-density decrease found at distance above 8 $R_E$ may be due to a higher rate of H ionization in the vicinity of the magnetopause because of increasing charge exchange interactions outside the magnetosphere.  It would be interesting if they could offer a plausible explanation for this increase.

Line 145-149: A more extensive explanation of the heritage and differences of the 2 models used for the data analysis would add to the significance of the reported results.

Line 218:220: An explanation, perhaps brief, of the theory would enhance the paper.

Line 244-245: This statement would be more meaningful if there were some "description" of the Chamberlain model.

Technical Corrections:
line 57: increase -> increases

Line 224-225: A reference to the 20% number would be appropriate.

Line 230: The "other studies" might be referenced or at least described.

Figure 4 has a description that refers to a "black line".  There are black squares but no black line.

Throughout the manuscript, separating introductory clauses from the main sentence by a comma would make reading the manuscript much easier, e.g., line 115, 120, etc.

---

## Author Comment (AC1)

Dear Prof. Perez,

many thanks from the authors for refereeing this manuscript and for the comments, which will help to improve this work.

Here is our response to your specific comments:

C1: Line 18-21 and Line 263-265:Authors suggest that the faster radial H-density decrease found at distance above 8 RE may be due to a higher rate of H ionization in the vicinity of the magnetopause because of increasing charge exchange interactions outside the magnetosphere.It would be interesting if they could offer a plausible explanation for this increase.

**to C1:**
We have added a possible explanation at the end of the manuscript (please see lines from 282).

C2: Line 145-149: A more extensive explanation of the heritage and differences of the 2 models used for the data analysis would add to the significance of the reported results.

**to C2:**
We have added some information about the models and a new reference with a very good overview of the chamberlain approach to the manuscript (please see lines from 162 and the new reference in the list [Beth et al., 2016]).

C3: Line 218:220: An explanation,perhaps brief,of the theory would enhance the paper.

**to C3:**
Please see also our response to C1, in particular the last sentence, at the end of the manuscript.

C4: Line 244-245:This statement would be more meaningful if there were some "description " of the Chamberlain model.

**to C4:**
"Chamberlain model" was changed to "1/R^3 model"

Technical Corrections:

line 57:increase ->increases
changed

Line 224-225:A reference to the 20%number would be appropriate.
Reference added

Line 230:The "other studies " might be referenced or at least described..
"Other studies" removed – reduced just to the 2 referenced studies

Figure 4 has a description that refers to a "black line ".There are black squares but no black line.
changed to "black squares"

Throughout the manuscript,separating introductory clauses from the main sentence by a comma would make reading the manuscript much easier,e.g.,line 115,120,etc.
Changed at the two positions

---

## Author Comment (AC2)

Dear Referee (#2),

many thanks from the authors for refereeing this manuscript and for the comments, which will help to improve this work.

Here is our response to your specific comments:

(A) For the correction of UVIS/HDAS observations, the authors determined the conversion factor, fc, for the emissions at r = 3.0-5.5 Re (overlapping region). The authors used a constant value of fc (3.285) for Analysis (3) to determine the Geocorona density profile for r = 3.0-15.0 Re. However, fc is not flat at r = 3.0-5.5 Re, and fc is ~3.1 at r > 8 Re. I suggest the authors use the r-dependent fc to determine the density profile. The different density profile found in Analysis (4) may be because the constant fc is applied.

**to (A):**
From its observational geometry (LOS impact distances = 3-6.2 Re) the validity of the TWINS-LAD nH-modelfits is between 3-8 Re geocentric distance ([Zoennchen et al., 2015]). For the fc-estimation it is reasonable to use only TWINS model values from inside this validity range 3-8 Re.

Distances directly covered by LOS's impacts have the lowest relative error in the density modelling. Therefore, with 3-5,5 Re we used a slightly reduced range with respect to the maximum possible. There is a TWINS-LAD reflection problem for LOS's with pointing near to the Sun (named in [Zoennchen et al., 2015]). Due to this, there is a lack of TWINS-LAD observations with LOS impact distances >5.5 Re at the dayside.

However, choosing the upper border a 8 Re or 5,5 Re would not make a significant difference to the average fc-value: 3-5,5 Re -> fc = 3.285; 3-8,0 Re -> fc= 3.22.

In the manuscript: We have added an info in the text (see red text line 192) and have removed the red line (TWINS calculated Ly-alpha column brightness) above 8 Re in Fig. 3 (A)+(B).

(B) The authors focus on r = 3.0-5.5 Re, for Analysis (2), but the TWINS-based model presented by Zoennchen et al. (2015) provides three-dimensional density profile as a function of local time, and latitudes, using harmonics expansion. The validity range of the model is 3 < r < 8 Re. Why do the authors rely on the TWINS-based model only at r = 3.0-5.5 Re? Is the model not valid for r > 5.5 Re on the dayside?

**to (B):**
The authors rely on the TWINS model inside its validity range 3-8 Re. In this distance range the radial dayside ecliptic density profile (GSE-longitudinal averaged between +/- 45° with respect to the solar direction) is in very good agreement with the density profile we found from UVIS/HDAC in Equ. (8) – see the hereafter attached plot.

It shows, that UVIS/HDAC confirms the TWINS profile between 3-8 Re (within 1% at 3 Re and 7.5% at 8 Re) and additionally provide an extended profile in the above distance range 8-15 Re.

[Figure]

(C) The authors used fc derived from the comparison between UVIS/HDAS and TWINS-based model, but the authors should be able to calculate fc from the comparison between UVIS/HDAS and the r^-3 model. I suggest they use the latter fc as well for Analyses (3) and (4).

**to (C):**
As visible in Fig. 3 (B) the fc-value from the r^-3 model is strongly varying between 3-8 Re from 2,37 (3 Re) to 3,08 (8 Re). This variation is from our point of view too large to assume it as a nearly constant calibration factor (therefore, we do not use it for the analysis). The fc-value from usage of the TWINS model is different, because it is varying only by 2-3 % around its average in this range.

(D) The authors assume that the exosphere r profile was similar for both UVIS/HDAS and TWINS cases due to comparable space weather conditions. However; (a) it is not adequately described in the manuscript how comparable the conditions are. I recommend the authors summarize the conditions in a table or something equivalent. (b) What is the advantage of using UVIS/HDAS on Cassini to model the density profile? TWINS observations are enough if the exosphere profile was similar and TWINS observations are better.

**to (D):**
(a) Values of the solar activity parameters sol. Ly-alpha and F10,7 and their possible min/max-ranges during solar cycle are added to the manuscript (see red text from line from 109 and from 120).

We compare the exosphere modelled during solar minimum (TWINS-model) with the exosphere (UVIS/HDAC) during a solar activity level, which is between minimum and medium. From our point of view the activity difference is small enough, that this comparison can be made. However, we see possible differences due to activity differences covered by the assumed error of +/- 25%.

(b) The aim of this manuscript is to show the extension of the radial exospheric density profile at the dayside from above 8 Re (which is the upper validity border of the TWINS model) up to 15 Re. This extended dayside distance region is very interesting, because it is crossing the magnetopause near the sub-solar point and is reaching the space outside the magnetosphere. Distances above 8 Re are out of the validity range of the TWINS model. The used UVIS/HDAC observations provide this opportunity.
In particular the best density value near the sub-solar point is important for the inversion of ENA- or Soft X-ray observations there.

(E) For Analysis (3), the fitting procedure/algorithm needs to be described. Also, the authors need to explain why they choose the model function.

**to (E):**
The density model function (Equ. 7) was choosen as identical to the mathematical form of the best fitting function (we found) for the background-free UVIS/HDAC-column brightness profile (Equ. 1).

For a LOS-integration inside the optically thin regime it can be expected, that the column brightness profile (outside the integral) and the density profile (inside the integral) have the same mathematical (powerlaw) form.

The LOS-integration itself is already good summarized in the referenced papers (Bailey & Gruntman, 2011, Zoennchen et al., 2013, 2015 etc.).

(F) It seems to me that the authors use r for two different parameters. One is the geocentric distance (the distance from the Earth center) used for the density models, and the other is the closest distance of the instrument line-of-sight from the Earth center (e.g., Figure 3) used for column brightness/density.

**to (F):**
The LOS impact distance to the Earth is the distance of the closest approach of the LOS to the Earth. It can be measured from Earth center (geocentric distance) or from Earth surface (altitude).

In case of the used UVIS/HDAC dayside observations the LOS impact distance is equal to the UVIS/HDAC geocentric distance itself. To be clear, we have changed the x-axis label in Figure (3) to "dayside geocentric UVIS/HDAC distance [Re]"

(G) For Analysis (1a), it is not adequately described what values of the coefficients of the TWINS-based model are used.

**to (G):**
We have added the number of the table in [Zoennchen et al., 2015], from which the parameter set was used for the TWINS-mode calculations (see red text in line 155)

(H) In addition to line-of-sight information, it is better to describe the field-of-view (FOV) information of UVIS/HDAS on Cassini. How wide is FOV? What is the pixel resolution (if the instrument can look at more than one direction)?

**to (H):**
The FOV=3° is already written in the manuscript – see in line 81.

Regarding other UVIS/HDAC instrumental information we added a reference to the "UVIS User's Guide" available at NASA PDS-atmosphere website (see red text in line 105)